# Heart Failure with Improved Ejection Fraction: Insight into the Variable Nature of Left Ventricular Systolic Function

**DOI:** 10.3390/ijerph192114400

**Published:** 2022-11-03

**Authors:** Maciej T. Wybraniec, Michał Orszulak, Klaudia Męcka, Katarzyna Mizia-Stec

**Affiliations:** 1First Department of Cardiology, School of Medicine in Katowice, Medical University of Silesia, 47 Ziołowa St., 40-635 Katowice, Poland; 2Upper-Silesian Medical Center, 40-635 Katowice, Poland; 3European Reference Network on Heart Diseases—ERN GUARD-HEART, 1105 AZ Amsterdam, The Netherlands

**Keywords:** heart failure with improved ejection fraction, HFimpEF, HFiEF, heart failure with recovered ejection fraction, HFrecEF

## Abstract

The progress of contemporary cardiovascular therapy has led to improved survival in patients with myocardial disease. However, the development of heart failure (HF) represents a common clinical challenge, regardless of the underlying myocardial pathology, due to the severely impaired quality of life and increased mortality comparable with malignant neoplasms. Left ventricular ejection fraction (LVEF) is the main index of systolic function and a key predictor of mortality among HF patients, hence its improvement represents the main indicator of response to instituted therapy. The introduction of complex pharmacotherapy for HF, increased availability of cardiac-implantable electronic devices and advances in the management of secondary causes of HF, including arrhythmia-induced cardiomyopathy, have led to significant increase in the proportion of patients with prominent improvement or even normalization of LVEF, paving the way for the identification of a new subgroup of HF with an improved ejection fraction (HFimpEF). Accumulating data has indicated that these patients share far better long-term prognoses than patients with stable or worsening LVEF. Due to diverse HF aetiology, the prevalence of HFimpEF ranges from roughly 10 to 40%, while the search for reliable predictors and genetic associations corresponding with this clinical presentation is under way. As contemporary guidelines focus mainly on the management of HF patients with clearly defined LVEF, the present review aimed to characterize the definition, epidemiology, predictors, clinical significance and principles of therapy of patients with HFimpEF.

## 1. Introduction

Heart failure (HF) represents a syndrome of symptoms resulting from multiple myocardial diseases leading to impaired cardiac output or increased intracardiac pressure at rest or during exertion [1]. It is estimated that the prevalence of HF is 1–2% of the adult population, reaching a value of about 20% in the population of patients over the age of 80. Irrespective of its diverse aetiology, once HF has developed, it confers an unfavourable outcome with a 5-year mortality rate ranging from 50% to nearly 70% [2,3]. Traditionally, HF is classified based on the left ventricular ejection fraction (LVEF), which constitutes its main prognostic factor, along with age, symptomatic class, level of natriuretic peptides, congestion status, number of acute decompensation of HF, hyponatremia, serum creatinine concentration and estimated glomerular filtration rate (eGFR), the presence of atrial fibrillation, diabetes mellitus and obesity [4,5,6]. Still, mortality is surprisingly high not only in HF with a reduced (HFrEF) and mildly reduced (HFmrEF) ejection fraction but also in patients with a preserved ejection fraction (HFpEF) [2,7]. Although, in general, patients with HFpEF share a better prognosis than those with HFrEF, the difference is negligible given the high prevalence of comorbidities in HFpEF [2]. The advances in pharmacotherapy of HF, the introduction of cardiac resynchronization therapy (CRT) and the improvement in the understanding of the reversible causes of HF have contributed to the increase in patients with a prominent improvement in LVEF [1]. More importantly, this subset of patients was associated with a more favourable outcome in contrast to stable or declining EF [8]. Surprisingly, an improved outcome has been confined mainly to patients with HFrEF, while the mortality of patients with a preserved EF has been more refractory to the improvements in therapy over time [7]. This has led to the identification of a new subgroup of HF with an improved (HFimpEF) or recovered ejection fraction (HFrecEF) [9,10]. Noteworthy is the fact that HFimpEF constitutes an indicator of a response to treatment and a clinical marker of favourable outcomes, as opposed to being a separate form of HF, given the miscellaneous aetiologies leading to HF [9]. However, the contemporary European Society of Cardiology Guidelines do not provide sufficient information on how to manage patients with an improvement in HF [1], while the American Heart Association Guidelines only deliver support for the maintenance of guidelines-directed medical therapy in patients with HFimpEF [10].

Thus, the present article sought to summarise the up-to-date knowledge on the definition, epidemiology, predictors, clinical significance and practical aspects of the management of patients with HFimpEF. For this purpose, the Medline and EmBase databases were queried to obtain original articles and review papers using the following set of keywords: heart failure with improved ejection fraction; HFimpEF; HFiEF; heart failure with recovered ejection fraction; HFrecEF; heart failure improvement; and heart failure change.

## 2. Definition of HFimpEF

Although the phenomenon of LVEF improvement has been reported for years, no universal definition of HFimpEF exists. The oldest and least precise, yet simple, definition is based on the cohort from the Val-HeFT trial, which required an initial LVEF measurement <35% and an increase in follow-up LVEF after 12 months to >40% [8]. An even simpler definition was provided by Jorgensen et al. in a recent meta-analysis [11], who defined HFimpEF as situation of an increase in LVEF ≥ 5% after a median time of 19 months [11]. Despite its simplicity, the usefulness of this definition is limited by the intra-observer variability in transthoracic echocardiography, which is similar to the presented threshold. Yet another approach was presented by the investigators engaged in the Swedish Heart Failure registry, which entailed an improvement in LVEF that met the requirement of an upgrade in HF subtype (HFrEF to HFmrEF or HFrEF or HFmrEF to HFpEF) [12]. This definition is also limited, as it may cover patients with only a slight improvement in the case of borderline LVEF. Another approach defines improvement as a recovery in LVEF from <35% to >50%, which is often referred to as HF with recovered EF (HFrecEF) [13]. This definition identifies patients with the best response; however, the term recovery may be imprecise as the state of improvement may be transient.

Still, the most contemporary definition comprises an initial LVEF < 40% and an increase of ≥10% and a follow-up measurement >40% [9]. This definition was adopted by the expert consensus published in 2020 and represents the most widely applied criterion of HFimpEF [9]. It was also recapitulated in the 2021 Universal Definition and Classification of Heart Failure [14] and the AHA guidelines on the management of HF [10].

### 2.1. Ejection Fraction Improvement: A Surrogate Marker of Reverse Remodelling

Although LVEF is regarded as the main categorizing parameter and marker of the response to treatment among HF patients, one should address the caveats related with its use as a universal therapeutic target. First, the symptoms of HF are secondary to impaired cardiac output related with both systolic and/or diastolic dysfunction. As obvious as it is, LVEF within the reference value does not exclude HF, and patients should be screened for structural abnormalities and elevated natriuretic peptides [1]. HFpEF patients, despite having a normal LVEF, are characterized by a high mortality rate and an impaired quality of life [2]. Second, LVEF is an echocardiographic parameter, which can be altered by inter- and intra-observer variability, and its variations may be triggered by human error or bias linked to the knowledge of a patient’s clinical condition. Third, in certain clinical settings, not LVEF but the change in echocardiographic volume parameters better reflects reverse LV remodelling. In patients with HFrEF and LV dyssynchrony, a decrease in the left ventricular end-systolic volume index (LVESVi) ≥15% from baseline at 6 months following cardiac resynchronization therapy device (CRT) implantation is regarded as a marker of therapeutic response, which may be accompanied by an increase in LVEF [15]. Reverse remodelling is defined as the normalization of LV geometry and reversal of alterations of the cellular and extracellular composition of the myocardium secondary to decreased mechanical and humoral stress [16]. Therefore, an increase in LVEF represents only a surrogate indicator of reverse remodelling. The improvement may also be reflected by a reduced sphericity index, defined as the ratio of the long-to-short axis of the LV both in systole and diastole, which was shown to normalize in patients with terminal HF subject to left-ventricular assist device (LVAD) implantation [17]. Fourth, LVEF is only one of plenty of parameters describing left ventricular systolic function, and accumulating data suggests that left ventricular global longitudinal strain (GLS) represents a more precise and stable parameter, with strong prognostic implications [18]. In a large cohort of patients, GLS was independently associated with all-cause mortality after adjustment for structural and functional abnormalities (LVEF) [15]. More importantly, baseline GLS adequately predicted the phenomenon of HFimpEF, suggesting that GLS might yield a more prognostic significance than a single measurement of LVEF [19]. Every 1% increase in GLS (less negative GLS) corresponded with a 10% higher odds ratio (OR) for HFimpEF defined as increase in LVEF to >40% from an initial LVEF ≤ 40% [19]. Although the clinical presentation of HFpEF is primarily linked to diastolic dysfunction and a high comorbidity burden, more data suggests that these patients experience a subclinical systolic dysfunction reflected by impaired GLS, which correlates with an increased level of natriuretic peptides but not with quality of life, nor symptomatic class [20]. Last but not least, the most desired response to treatment by patients is the reduction in symptoms. The improvement in LVEF may not be accompanied by a shift in symptomatic class and, conversely, symptomatic improvement may coexist with stable LVEF [5,6]. A recent study by Wohlfahrt et al. showed that a recovery of LVEF to >50% from <35% was associated with a significant increase in patient-reported quality of life and the Kansas City Cardiomyopathy Questionnaire score, as well as functional status [21]. All in all, LVEF fluctuation seems to be the easiest and most reliable indicator of the general HF trajectory and its response to treatment.

### 2.2. Prevalence and Predictors of LVEF Improvement

The rate of HFimpEF among the population of patients with HF depends on the applied definition, the proportion of patients with different aetiologies, the reversible causes of HF and the intensity and appropriateness of HF therapy. In the Val-HeFT trial, the criteria of improvement were met in 9.1% of 3519 patients with a baseline LVEF < 35% [8]. The longitudinal analysis of the trajectory of LVEF variations by Savarese et al. based on the Swedish Heart Failure Registry showed that 26% of patients with a baseline HFrEF and 25% of patients with an initial HFmrEF improved to a better systolic function subtype of HF [12]. In a recent study by Su and co-workers, HFimpEF, defined as increase in LVEF from ≥10% to >40%, occurred in 18% of patients [22]. In a recent study by Li et al., HFimpEF, defined as an absolute increase in LVEF by 10% (regardless of baseline value), was found in 41.2% of cases [23]. In a recent meta-analysis by He et al. performed on 9491 patients from nine studies, HFimpEF was present in 22.6% of patients [24]. All in all, the prevalence of HFimpEF varies from roughly 10 to 40% [9].

Given the distinct phenotype, it seems reasonable to pursue reliable predictors of systolic function improvement among the broad population of patients with HF (Table 1). Hitherto, a report by Savarese delivered evidence that a scenario of HFimpEF is more likely among females (the highest OR of 1.76, 95%CI: 1.47–2.12), in patients with a non-ischemic aetiology of HF, patients with a history of atrial fibrillation/atrial flutter, outpatient management with no HF exacerbations and patients with arterial hypertension, anaemia and a higher social and financial status [12]. These findings highlighted the importance of the reversible causes of HF, such as tachycardia-induced cardiomyopathy in the course of AF, which, if properly handled, can lead to an improvement in or even the normalization of LV systolic function. A meta-analysis by Jorgensen et al. also delivered evidence that a lower baseline LV end-diastolic diameter (LVEDD) is associated with a greater rate of HFimpEF [11]. Other studies identified a younger age, beta-blocker use, a valid indication for pulmonary vein isolation or CRT implantation, as well as chemotherapy cardiotoxicity among patients with malignant neoplasm, troponin levels within the reference value and elevated natriuretic peptides as independent predictors of HFimpEF [23,24,25,26]. In addition, convincing data suggests that baseline GLS accurately predicts future variations in LVEF, indicating that GLS is a more reliable and stable in-time parameter of systolic function [16]. Studies concerning the baseline use of cardiac magnetic resonance imaging (CMR) have demonstrated that the absence of late gadolinium enhancement in patients with non-ischaemic cardiomyopathy heralds a favourable response to pharmacotherapy and a recovery in LVEF [27].

### 2.3. HFimpEF and Favourable Prognosis

The main premise for identifying responders to HF treatment consists of the strong evidence of a far better prognosis in this group of HF patients. Despite numerous predictors of HF outcome, such as LVEF, symptomatic class, frequent hospitalization for decompensated HF or the level of natriuretic peptides, the prognosis of HF remains vastly unfavourable and uncertain. The phenomenon of LVEF improvement identifies a subset of patients with a distinct set of phenotypes from patients with stable or declining LVEF [13], which is linked to better survival than patients with stable or declining LVEF [8,11,12,13,22,23,24,26].

Jorgensen et al. found that patients with an increase in LVEF ≥5% within a median time of follow-up of 19 months had a lower risk of death than patients with persistently reduced LVEF (5.8% vs. 17.5%, HR 0.34; 95%CI: 0.28–0.41, *p* < 0.001) [11]. Accordingly, Savarese and co-workers found that a transition to a better systolic-function group was linked to a significantly lower risk of a composite endpoint of death or hospitalization for HF than patients with stable LVEF (37% vs. 59%, HR 0.62, 95%CI: 0.55–0.69) [12]. These results comply with recent observational studies from Asia, which also showed a lower all-cause mortality and risk of hospitalization for HF among patients with HFimpEF [22,23]. These findings were also summarized in the meta-analysis by He et al. comprising nine studies with 9491 participants, which delivered evidence for a significantly lower risk of hospitalization for HF and death among HFimpEF patients in comparison to both HFrEF (mortality: HR: 0.44, 95% CI: 0.33–0.60; hospitalization for HF: HR: 0.40, 95% CI: 0.20–0.82) and, surprisingly, HFpEF patients (mortality: HR: 0.42, 95% CI: 0.32–0.55; hospitalization for HF: HR: 0.73, 95% CI: 0.58–0.92) [24].

It is vital to note that HFimpEF does not represent a separate form of HF, rather a group of phenotypes with different aetiologies that share a better prognosis than patients without LVEF improvement [13]. Still, within the HFimpEF group, patients can also be stratified into different subgroups depending on their long-term prognosis and probability of sustained LVEF improvement [13]. Patients and their families should be made familiar with this fact and should be subject to a meticulous clinical and echocardiographic follow-up.

### 2.4. Pathophysiology of Reverse Left Ventricular Remodelling

The pathophysiology of a failing heart represents a complex interplay concerning alterations in gene expression, changes in metabolic pathways and the contents of the extracellular matrix. The progression of HF leads to modifications of the expression of multiple genes, including immune-modulation genes [28,29], which persists in more than 75% of patients despite an overt clinical and symptomatic improvement related with the use of a left ventricular assist device (LVAD) [28]. This suggests that, despite LVEF recovery, alterations in gene transcription persist and may facilitate future recurrence [28]. Once shifted, the transcription of genes promotes HF relapse.

The main metabolic change within cardiomyocytes in HF is the decreased uptake of fatty acids by cardiomyocytes and its reduced beta-oxidation and shift towards glycolytic pathways, which translates into an increased intracellular concentration of cytosolic lactate rather than pyruvate [30]. In addition, the chronic activation of sympathetic nervous systems and the activation of beta-adrenergic receptors leads to further adverse cardiac remodelling and a limited adrenergic reserve [31,32]. Of note is the fact that, in HF, the response to beta-2 adrenergic receptors is suppressed, while the response to sympathetic stimulation is mediated primarily via the beta-1-receptor [32]. So far, different therapies for HF, including LVAD, have had little or no impact on the reversal of these alterations with persistent low mitochondrial oxidative capacity [33]. Certain data suggests that mineralocorticoid receptor antagonists (MRA) can yield a protective impact on mitochondrial function and cellular energetics [34].

Dysregulation in the intra-cellular levels of calcium and sodium plays a key role in the electrophysiologic abnormalities of a failing heart [35]. Cardiomyocytes in HF have a higher Ca^2+^ influx and a reduced uptake of CA^2+^ to the sarcoplasmic reticulum due to the dysfunction of energy-dependent SERCA2 [35], which triggers a higher diastolic level of cytosolic Ca^2+^, the spontaneous releases of Ca^2+^ from the sarcoplasmic reticulum and an increased risk of afterdepolarizations and life-threatening ventricular arrhythmia [35]. The decreased density of the T-tubular network leads to impaired excitation–contraction coupling [35]. An increased late sodium current causes a prolonged repolarization time, which can also contribute to an increased QTc and a risk of afterdepolarizations [35]. It is vital to note that beta-blockers have been shown to induce the reversal of myofibrillar remodelling [36] and improve the real-time calcium-dependent energy consumption by cardiomyocytes [37].

On the other hand, the reversal of pathologic changes has been demonstrated in the extracellular matrix (ECM), which is a vital habitat for cardiomyocytes necessary for its structural support and neurohormonal signalling. The environment of cardiomyocytes in HF is characterized by an increased level of the cross-linked fraction of collagen, osteonectin, osteopontin, tenascin C, thrombospondin, periostin and matrix metalloproteinases and a decreased activity of tissue inhibitors of metalloproteinases (TIMPs) [38]. The clinical use of inhibitors of the renin–angiotensin–aldosterone system has been linked to the reversal of some of the alterations of ECM, particularly the decrease in the total contents of collagen and its cross-linked fraction, translating into decreased fibrosis, stiffness and a lower left ventricular mass [38]. This has been demonstrated particularly with reference to angiotensin-converting enzyme inhibitors (ACEI) in the prevention and treatment of ischemic and non-ischemic HF [39] and MRAs [34,40], and was even more pronounced in patients receiving angiotensin receptor neprilysin inhibitors (ARNI) [41,42,43].

All in all, reverse remodelling reflected by the clinical setting of HFimpEF is accompanied by a reversal of pathology within ECM; however, an abnormal metabolism of cardiomyocytes and altered gene regulation warrant caution when it comes to the cessation of effective treatment.

## 3. General Therapy of Heart Failure Irrespective of Aetiology

The initial management of HF comprises symptomatic relief mediated in part by diuretic therapy, the institution of disease-modifying pharmacotherapy and establishing the aetiology of HF. Irrespective of aetiology, all patients with HFrEF should receive the recommended guideline-directed medical therapy (GDMT), which has been shown to reduce the risk of hospitalization for HF and to prolong survival [1,10]. According to the current European Guidelines on the management of HF [1], an array of disease-modifying drugs including beta-blockers, ACEI, ARNI, MRA, ivabradine and sodium–glucose cotransporter-2 (SGLT2) inhibitors, are recommended in patients with HFrEF in the light of strong scientific evidence from landmark randomized controlled trials. However, their application may be considered in patients with HFmrEF, as this represents a sound extension of indications based on expert consensus and data from surrogate studies, but the evidence regarding an improvement in outcome is lacking. In the setting of HFpEF, so far, only SGLTi has been shown to modify survival, while guidelines concentrate on the management of abundant comorbidities accompanying HFpEF [1].

Less is known how these disease-modifying medications promote functional and structural reverse remodelling, such as a reduction in LVESV, reduction in the level of sphericity index and, most importantly, an increase in LVEF.

The impact of beta-blockade on reverse remodelling was demonstrated by Hall and co-workers, who gave evidence for the long-term increase in LVEF and the decrease in left ventricular mass and index of sphericity, following the initial negative inotropic effect resulting in the transient decrease in LVEF in the first days of treatment [44]. Beta-blockers were shown to exert a beneficial effect on myocardial energetics reflected by an improvement in minute work without an increase in oxygen consumption, regardless of the aetiology of HF [45,46].

The cornerstone of modern HF therapy is based on the use of ACEI, which has been shown to slow down ventricular remodelling and neurohumoral activation in HF of ischemic aetiology [47]. The application of ACEI triggers the reversal of cardiomyocyte hypertrophy and myocardial fibrosis [48]. The addition of angiotensin-receptor blockers (ARB) to ACEI or beta-blockers was associated with a further reduction in the left ventricular diastolic diameter and an increase in LVEF [49]. Although ARB should not be used together with ACEI on account of the risk of hyperkalaemia, one should consider using ARB in HF patients who are intolerant of ACEI [1,49].

Further blockade of renin–angiotensin–aldosterone syndrome with the use of MRA has been linked not only to a reduction in the risk of death and hospitalization for HF among HFrEF patients but also to antifibrotic effects within the myocardium and a resultant reduction in arrhythmogenesis [50]. In a cohort of patients with post-infarction HFrEF, the randomized use of spironolactone on top of HF pharmacotherapy was associated with a more profound increase in LVEF, the suppression of the LV end-diastolic volume index increase and a lower concentration of the plasma procollagen type III aminoterminal peptide level in comparison to the non-MRA cohort [51].

Accumulating evidence has shed light on the role of sacubitril/valsartan, which not only acts as ARB but also blocks the dipeptidyl peptidase IV responsible for the turnover of bradykinin and natriuretic peptides. Along with its impact on the reduction in the risk of death and hospitalization for HF among HFrEF patients in comparison to ACEI [52], sacubitril/valsartan promotes a marked increase in LVEF and a decrease in LVESV and left ventricular end-diastolic volume (LVEDV), as well as exerts an antiarrhythmic effect on LV [53]. A recent study suggests that its effect on reverse remodelling might be more pronounced in non-ischemic cardiomyopathy than among HF secondary to ischaemia [54].

Yet another advance in the pharmacotherapy of HF was the introduction of SGLT2 inhibitors as independent drugs for the therapy of HF, irrespective of the diagnosis of diabetes mellitus [55]. The current evidence suggests that their beneficial effect is valid not only for patients with HFrEF but also for patients with HFpEF [56,57]. Recently published results of the Deliver Study included patients with HFmrEF and HFpEF, including patients with a prior LVEF < 40% who were consistent with the HFimpEF definition [57]. This was the first randomized controlled trial to investigate any pharmacological agent in this subset of patients and it showed that the efficacy of dapagliflozin in the reduction in primary endpoint was even more profound in HFimpEF patients than in the rest of the HFmrEF and HFpEF population [57]. SGLT2 inhibitors are characterized by their antifibrotic and anti-inflammatory and diuretic effect, which translates into a significant reversal of pathologic echocardiographic alterations typical of HF [58]. A recent meta-analysis by Theofilis et al. covering 2351 patients denoted that the application of SGLT2 inhibitors triggered an increase in LVEF and GLS and a decrease in LVESV, the left ventricular mass index, the left atrial volume index and the E/e’ index [58].

## 4. Management of Patients with HFimpEF: Pharmacotherapy and Surveillance

The chronic character of HF raises an important clinical aspect in terms of the management of HF patients concerning the continuation of GDMT and limiting the burden for healthcare system in patients with LVEF improvement. In addition, the majority of patients perceive their health and quality of life by the number of drugs that they are recommended to take daily. The question of the maintenance of pharmacotherapy in patients with improvement or even full recovery of LVEF and the relief of symptoms is an important issue for patients that was addressed in a randomized TRED-HF study [59]. This open-label study enrolled 51 patients with dilated cardiomyopathy with a full recovery of LVEF from <40% to ≥50% and a normalization of LVEDV and natriuretic peptides level [59], who were randomized in 1:1 ratio to a stepped withdrawal of drugs or a continuation of treatment [59]. The study showed that phased withdrawal led to a relapse of HF defined as a decrease in LVEF >10% or HF symptoms in 44% of patients in comparison to none in the control group [59]. This study delivered the first evidence that the recovery of LVEF is dependent on instituted therapy and thus effective pharmacotherapy should be maintained. Noteworthy is the fact that HFimpEF should rather be designated as the transient remission of systolic dysfunction since the recurrent deterioration of LVEF is a frequent phenomenon. It was documented that among patients with a non-ischemic aetiology of HF, a reoccurrence of systolic dysfunction was reported in nearly 19% of all cases and was associated with the cessation of HF medications [60]. This has paved way for a single recommendation concerning HFimpEF of Class I level B in the current AHA Guidelines on the management of HF for maintaining the GDMT that led to the improvement in systolic function (Figure 1) [10]. Following initial intensive diuretic therapy in patients with a decompensation of HF, the dose of diuretics may be gradually reduced according to the volemic status, yet it should not be completely withdrawn upon discharge. The authors reckon that the maintenance of a low dose of diuretics is a wise option in patients with severe systolic dysfunction in order to prevent future HF decompensations.

The surveillance of patients with HFimpEF should comprise medical consultation with a physical examination, ECG and natriuretic peptides screening for left bundle branch block every 6 months until 12–18 months of HFimpEF and subsequently every 6–12 months [9]. Patients should be assessed in terms of diuretic therapy. Transthoracic echocardiography should be performed every 6 months until 12–18 months of HFimpEF and repeated every 6–12 months afterwards [9]. It is advisable to perform cardiac magnetic resonance after 1 year of clinically stable HFimpEF in order to assess the degree of fibrosis and to perform genetic testing for the diagnosis of dilated cardiomyopathy to assess the likelihood of an improvement in systolic function and the risk of sudden cardiac death [9]. Truncating mutations within titin genes (tTTN) are generally linked to a more favourable course with a higher rate of HFimpEF achieved by means of GDMT [61]. On the other hand, mutations in lamin genes (LMNA), FLNC, SCN5A and DSP genes were associated a high risk of sudden cardiac death despite an overt improvement in systolic function following treatment administration [62]. Such profiling of HF patients may help plan the right schedule of follow-up visits and inform them about the expected prognosis. The present approach is summarized in Figure 1.

## 5. Cardiac Implantable Electronic Device and HFimpEF

The issue of HF therapy in the context of HFimpEF also concerns cardiac implantable electronic devices (CIED). This complex aspect of HF therapy comprises: (1) the choice of an optimal population of patients who will respond to cardiac resynchronization therapy (CRT) in patients with HFrEF and the concomitant electrocardiographic signs of left ventricular dyssynchrony; (2) the consideration of elective implantable cardioverter–defibrillator (ICD)/CRT implantation and the choice of its right timing in the context of potential systolic function improvement; and (3) the reassessment of indications for CIED replacement in patients who already have undergone implantation and require its elective replacement.

### 5.1. ICD and HFimpEF

The contemporary guidelines recommend the implantation of an ICD in the primary prevention of sudden cardiac death in HFrEF with LVEF ≤35% in symptomatic New York Heart Association class 2–3 who are expected to survive >1 year in good clinical condition, following 3 months of optimal medical therapy in order to reduce the risk of sudden cardiac death and all-cause mortality [1]. According to the results of the DANISH Study, the benefits of ICD are greater in an ischemic rather than a non-ischemic aetiology of HF [63].

HFrEF pharmacotherapy should be implemented and continued for an implicit period of time before determining the need for device therapies [1,64]. ICD implantation is recommended only if a minimum of 3 months of optimal medical therapy has failed to increase the LVEF to >35%. Depending on clinical status, the decision to conduct follow-up imaging might be shorter or longer depending on the risk for sudden cardiac death. Over the past few years, the advent of two medications, namely SGLT2 inhibitors and ARNI, became the gold standard in HF treatment. The introduction of newer agents complicates the timing for ICD implantation due to the uncertainty about their time of action and dealing with up-titration. Therefore, Garcia et al. [65] suggest 9 months for achieving optimal medical therapy, as ICD implantation may become redundant owing to systolic function improvement [66]. In particular, the application of sacubitril/valsartan obviated the need for ICD implantation in 25% of patients in the SAVE-ICD study, especially in non-ischemic aetiology [67] and even in 60% of patients in the study by Pastore et al. [68]. The question remains whether an improvement in LVEF translates into a reduced risk of sudden cardiac death. The meta-analysis by Smer and co-workers comprising 3959 patients provided evidence that an improvement in LVEF > 35% in comparison to persistent systolic dysfunction is associated with a lower risk of adequate ICD therapy (3.3%/year vs. 7.2%/year, RR 0.52, *p* < 0.001) [69].

One can assume that it is the type of cardiomyopathy that determines the time to reassess LV function, hence in DCM, peripartum cardiomyopathy, tachycardia-induced cardiomyopathy and myocarditis, a waiting period of 6 months may be required to allow adequate reverse remodelling before re-evaluating the indications for ICD implantation, while improvements in ischemic HF are rare beyond 3 months of GDMT [70,71].

### 5.2. CRT and HFimpEF

The efficacy of CRT is commonly treated as being a ‘CRT responder’, which may fall into a particular category of HFimpEF. The indications for CRT encompass symptomatic patients with LVEF < 35% and QRS prolongation ≥ 130 ms, optimally ≥ 150 ms, with left bundle branch morphology [1]. The ideal features of a CRT responder are highlighted in Table 1. The most common definition of CRT response is that the patient has fewer symptoms and/or better clinical outcomes with this therapy than without it. Interestingly, the most commonly used criterion for CRT efficacy is not LVEF improvement (such as in the assessment of HFimpEF) but LVESV reduction by >15%, because it corelates more accurately with long-term survival in CRT recipients [72,73]. Moreover, there is no defined time frame for CRT assessment. In the majority of trials, CRT evaluation has been performed after several (up to twelve) months after implantation. This time period is crucial for the determination of being a CRT responder. After this period, further measurable CRT benefits are not observed, but the primordial positive effect of resynchronization therapy is long-lasting. Nagase et al. [74] found that 4 years after CRT device replacement (due to battery depletion), no changes in LVESV/LVEF were observed, but the initial echocardiographic response predicts the subsequent very-long-term prognosis.

### 5.3. Maintenance of CIED in HFimpEF Patients

The significance of CRT and its long-term maintenance as a therapy is indisputable because an interruption of biventricular pacing leads to a worsening of LV function and an increase in functional mitral regurgitation [75,76]. The management of ICD for the primary prevention of SCD among HFimpEF patients remains a clinical challenge, especially when battery depletion is approaching and a decision about generator replacement must be taken. Approximately 25% of patients with implanted ICD experience LVEF improvement and meet the criteria of HFimpEF. In the PROSE-ICD study [77], the incidence rate for appropriate ICD shock per 100 patient-years was 5.5, 2.4 and 1.7% for LVEF < 35%, 36–54% and LVEF > 55%, respectively. In a meta-analysis of 16 studies, patients with HFimpEF had half the risk of ICD-rendered therapy compared to the HFrEF group [69]. LVEF improvement is associated with a decreased risk of ventricular tachycardia, however there is still a persistent arrhythmic risk among recovered EF patients, with a 3.3% per year rate of appropriate ICD therapy among those with LVEF ≥ 45% [69], while in comparison, the risk of ICD therapy among patients with primary SCD prevention reaches 22.9% during a roughly 40-months follow-up. ICD is effective for a reduction in SCD, but we must bear in mind that the proportion of ICD interventions in the follow-up cannot be used as a surrogate for its efficacy in preventing mortality.

An analysis of the SCD-HeFT trial [78] (Sudden Cardiac Death in Heart Failure Trial) showed that patients who had an improvement in EF to >35% during follow-up accrued a similar mortality benefit with an ICD as those whose EF remained ≤35%, but in this analysis the definition of HFimpEF was used differently than nowadays. Randomized controlled trials studying whether ‘to replace ICD generator in HFimpEF or not’ are strongly needed. Therefore, in the absence of supportive data, it seems reasonable to pursue active ICD therapy among HFimpEF patients.

## 6. Functional Mitral Valve Insufficiency: A Common Pitfall in the Natural History of Heart Failure

A common aftermath of left ventricular remodelling is the dilatation of the mitral valve annulus and the displacement of the apical and lateral papillary muscles causing the tethering of mitral leaflets and a lack of mitral coaptation, which leads to functional mitral regurgitation (fMR) without the structural abnormalities of leaflets and valve apparatus [79,80]. The degree of fMR is variable based on LV dilatation and the aetiology of HF, but it accelerates the further enlargement of LV and the decline in LVEF, which in turn aggravates the level of mitral regurgitation in a vicious circle. This phenomenon is typical for a severe enlargement of LV both in ischemic and non-ischemic HF, as well as in long-standing AF, which causes a dilatation of the left atrium and mitral annulus leading to fMR without an enlargement of LV and systolic dysfunction [80]. The approach to the treatment of severe fMR should concentrate primarily on HF pharmacotherapy and biventricular pacing with CRT [81], which have both been shown to limit the extent of fMR and improve symptoms. The use of sacubitril/valsartan in patients with heart failure and fMR was linked to a reduced effective regurgitant orifice area in comparison to valsartan alone [82].

If symptoms persist despite GDMT and CRT implantation, patients should be assessed by a heart team in terms of eligibility for surgery [79]. MV repair or replacement for fMR is generally recommended in case of concomitant indications for CABG or other cardiac surgery [79]. In patients ineligible for surgery based on high operative risk or a lack of indications for revascularization, a minimally invasive approach with the use of transcatheter edge-to-edge repair may be considered [79]. The data on the results of surgical interventions for fMR showed that the mean LVESVi decreased at a 1- and 2-year follow-up following, both in terms of mitral valve replacement and repair, in comparison to conservative treatment [83,84]. Still, this reverse remodelling was not accompanied by any survival benefit [84]. Of note is the fact that the recurrence of significant fMR at a 2-year follow-up was significantly higher among patients undergoing a repair rather than a replacement of the mitral valve [84].

There is dispute over the role of transcatheter edge-to-edge repair of the mitral valve in the treatment of fMR given the conflicting results of the MITRA-FR and COAPT trials [85,86]. The COAPT study showed a significant reduction in the left ventricular end-diastolic volume (LVEDV) from baseline in the device group and a significant reduction in hospitalization for HF (HR 0.53, 95%CI: 0.40–0.70) and death from any cause (HR 0.62, 95%CI: 0.46–0.82) at a 24-month follow-up [85]. Conversely, the MITRA-FR study did not show any difference in terms of the rate of composite endpoint of hospitalizations for HF and all-cause death between the device and control groups [86]. As the results may be related with the disparity between the severity of fMR and LVEF, the impact of transcatheter mitral valve repair on reverse remodelling is undisputable. Gripari et al. found that percutaneous mitral valve repair leads to a significant reduction in both LVESVi and LVEDVi and an increase in LVEF at 30 days and 6 months following the procedure [87].

### HFimpEF in the Context of Specific Aetiologies of Heart Failure

The course of HF is inextricably related with its aetiology, hence the identification of HF cause and the prompt institution of specific treatment on top of GDMT represent the vital steps in the management of HF (Figure 1, Table 2). Ischaemic heart disease, most importantly coronary artery disease, represents the dominant aetiology of HF. It is generally thought that patients with an ischaemic aetiology of systolic dysfunction have a lower chance of LVEF improvement than patients with a non-ischemic cardiomyopathy. Routine viability testing may help to identify patients who will benefit from surgical revascularization in terms of LVEF improvement, but it does not improve mortality [88]. In the case of ischaemic aetiology, surgical revascularization with coronary artery bypass grafting (CABG) represents a legitimate approach given the results of the STICHES trial [89]. Although randomized data on the direct comparison between CABG and PCI in HFrEF is lacking, a retrospective study covering 12,113 patients performed in Ontario demonstrated that patients undergoing PCI had a significantly higher mortality (OR 1.6 95%CI: 1.3–1.7) and rate of major adverse cardiovascular events than patients treated with surgical revascularization [90]. In the recent REVIVED-BCIS2 trial, PCI in patients with LVEF ≤ 35% failed to show any benefit in terms of a reduction in mortality or rate of hospitalizations for HF in comparison to pharmacotherapy, neither did it translate into an improved LVEF [91]. It should be noted, however, that a considerable proportion of patients with HFrEF and a high operative risk may benefit from PCI if feasible [92].

Clinicians should pay particular attention to non-ischemic aetiologies of HFrEF, which give a greater promise of reversal of systolic dysfunction. Dilated cardiomyopathy represents a heterogenous group of primary disorders of myocardium, which are related with mutations in sarcomere genes, myocarditis, auto-immune response following myocarditis, toxic injury or thyroid dysfunction. Given a nearly 35% prevalence of familial DCM, genetic testing should be utilized in order to identify the mutations responsible for the clinical phenotype [93]. Truncating mutations in titin genes have been shown to correspond with a favourable clinical outcome and response to pharmacotherapy [61].

Tachyarrhythmia, most commonly due to atrial fibrillation and atrial flutter, can be both a cause and sequelae of HF [94]. In the context of HFimpEF, the institution of rate and rhythm control may yield a recovery of systolic function in patients with tachycardia-induced cardiomyopathy (Table 2). In patients with AF overlapping advanced HFrEF, a return of the sinus rhythm may contribute to an improvement in LVEF [95]. In the case of tachyarrhythmia-induced cardiomyopathy, the recovery of LVEF up to 6 months confirms the right diagnosis and heralds a good prognosis.
ijerph-19-14400-t002_Table 2Table 2Clinical features of ischemic and selected non-ischemic aetiologies of heart failure with respect to the occurrence of heart failure with improved ejection fraction.HF AetiologySpecific AetiologySpecific Treatment on Top of GDMT/CRT/Treatment of fMRProbability of HFimpEFClinical FeaturesRefs.Ischaemic MI-related myocardial scar with further remodellingMyocardial revascularization in patients with viable myocardiumPreference for CABG vs. PCI in patients with HFrEFLowThe most frequent aetiology of HFrEFIschaemic aetiology is generally associated with a worse prognosis than non-ischaemic aetiologyCABG in HFrEF is related with survival benefit and a lower risk of recurrent myocardial infarction and revascularization than PCICABG and PCI may promote reverse remodelling with a significant increase in LVEFViability testing prior to revascularization predicts HFimpEF but does not predict clinical outcome[88,89,90]Global ischemia/myocardial stunning/freezingHighDilated cardiomyopathyFamilial/sporadic DCMIn selected cases of inflammatory viral-negative myocarditis, immunosuppressive treatment may be beneficial RCT are ongoingModerate-to-highDilatation of LV and systolic dysfunction without significant lesions in coronary arteries and a lack of overt secondary causes of HFrEFVariable outcome and chance of HFimpEF depending on genetic variants: tTTN genes mutations predict a good response to pharmacological treatmentActive viral myocarditis steroid/immunosuppressive treatment is contraindicated except for Loeffler and Giant Cell myocarditisInflammatory, viral-negative DCM immunosuppressive treatment may be beneficial[91,93,96,97,98,99,100]Inflammatory DCMActive myocarditisArrhythmia-induced cardiomyopathyAF/AFl/ATRate control (BBs, CaB, digoxin)Rhythm control:
(a)AAD(b)PVI
In refractory HFrEF with atrial tachyarrhythmia: AV junction ablation and CRT implantationHighPersistent HR > 100 bpmIschaemic aetiology is excludedNo other obvious aetiology of HF, e.g., alcohol abuse, uncontrolled arterial hypertensionAbsence of LV hypertrophyNormal or only mildly increased size of LVRemission of LV systolic dysfunction following successful rate or rhythm control within 1–6 monthsRecurrence of LV systolic dysfunction after new onset arrhythmia or impaired rate controlIn AF and HFrEF, evidence for a reduction in death from any cause or hospitalization for HF in patients subject to PVI vs. pharmacotherapy alone[94,95]PVB/nsVTAADVentricular ablationHighPacing-induced cardiomyopathyPromotion of native AV conductionReduction in RV pacingHis bundle pacingHighVentricular dyssynchrony: LBBBBiventricular pacing with CRTHighValvular heart diseaseASAortic valve replacement/TAVIHigh to low depending on stageSystolic dysfunction remains a key indicator of disease progression and an indication of intervention in asymptomatic valvular heart diseaseIn AS, systolic dysfunction may be the cause of low-flow phenomenon and a lack of diagnostic gradients across the valveIn low-flow AS, dobutamine stress echocardiography is indicated to assess the presence of flow reserve, which is a ≥20% increase in stroke volume in response to low-dose dobutamine and excludes pseudosevere ASIn HFrEF of other aetiologies, the presence of even moderate valvular heart disease may contribute to clinical condition[79,101,102]ARAortic valve replacement or repairPrimary MRMitral valve repair/replacementMSPercutaneous mitral commissurotomySurgical commissurotomyMitral valve replacementChemotherapy cardiotoxicityType 1—anthracycline-likeWithdrawal of chemotherapy/immunotherapy or change in therapy regimenCardioprotective treatment consistent with GDMT: BBs and ACEILowType 1 cardiotoxicity is characterized by late onset (years) and persistent impairment of systolic functionType 2 cardiotoxicity occurs directly during and after the initiation of treatment (weeks) and systolic dysfunction is reversible following cessation of treatment[103]Type 2Trastuzumab-likeHighACEI—angiotensin-converting enzyme inhibitors; AF—atrial fibrillation; AV—atrioventricular; BBs—beta-blockers, CaB—calcium channel blockers; CABG—coronary artery bypass grafting; CRT—cardiac resynchronisation therapy; DCM—dilated cardiomyopathy; HFimpEF—heart failure with improved ejection fraction; HFrEF—heart failure with reduced ejection fraction; GDMT—guidelines-directed medical therapy; HR—heart rate; LV—left ventricle; RV—right ventricular bpm—beats per minute; nsVT—non-sustained ventricular tachycardia; AS—aortic valve stenosis; AR—aortic valve regurgitation; MS—mitral valve stenosis; MR—mitral valve regurgitation; LBBB—left bundle branch block; PCI—percutaneous coronary intervention; PVC—premature ventricular contraction; PVI—pulmonary vein isolation; TAVI—transcatheter aortic valve implantation; tTTN—truncating titin genes. 


## 7. Conclusions

Due to the heterogenous aetiology of HF, no universal definition of the state of improvement exists. The most widespread criteria include an initial LVEF < 40% with a further absolute increase in LVEF of ≥10% above the threshold of 40%. This condition of LVEF recovery is present in 10–40% of patients diagnosed with HF. One should rather use the term ‘transient remission of systolic dysfunction’, as LVEF is variable and recurrent LVEF deterioration may occur. Patients with HFimpEF are characterized by a 50% lower risk of death and/or hospitalization of HF in contrast to patients with HFrEF or HFpEF. This underscores the importance of the identification of this subset of HF patients, given the generally ominous prognosis of the HF population.

## Figures and Tables

**Figure 1 ijerph-19-14400-f001:**
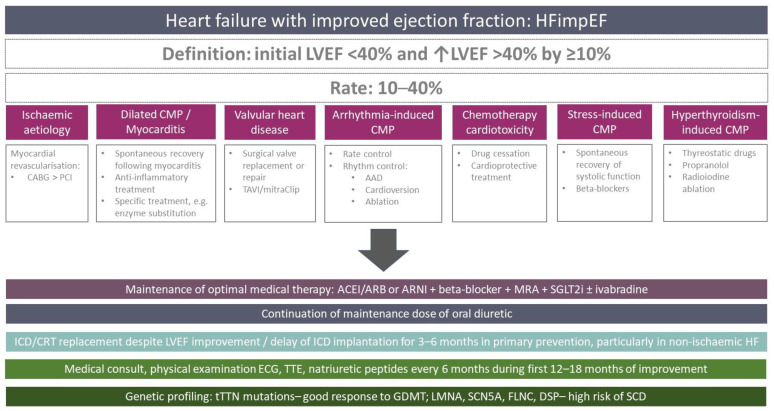
Overview of different clinical scenarios of heart failure with improved ejection fraction [9,10]. ACEI—angiotensin-converting enzyme inhibitors; ARB—angiotensin receptor blockers; CMP—cardiomyopathy; CRT—cardiac resynchronisation therapy; GDMT—guidelines-directed medical therapy; HF—heart failure; HFimpEF—heart failure with improved ejection fraction; PCI—percutaneous coronary intervention; CABG—coronary artery bypass grafting; LVEF—left ventricular ejection fraction; MRA—mineralocorticoid receptor antagonists; SGLT-2i—sodium–glucose cotransporter 2 inhibitors; SCD—sudden cardiac death; ECG—electrocardiographic study; TTE—transthoracic echocardiography; SCN5A—sodium voltage-gated channel alpha subunit 5; tTTN—truncating titin genes; FLNC—filamin C gene; DSP—desmoplakin gene; TAVI—transcatheter aortic valve implantation.

**Table 1 ijerph-19-14400-t001:** Baseline predictors of heart failure with improved ejection fraction [11,12,16].

Baseline Predictors of HFimpEF	CRT Response Characteristics
Non-ischaemic aetiology of HF Female sex ↓Age ↓LVEDD ↑Blood pressure Beta-blocker use AF/AFl with TIC Pulmonary vein isolation CRT implantation Anaemia Troponin level within reference value Higher social and financial status Lack of HF exacerbations ↑LV GLS Lack of LGE on CMR Shorter duration of HF Lack of LBBB	Non-ischaemic aetiology of HF Female sex QRS complex width ≥130 ms, optimal ≥150 ms LBBB > non-LBBB Prolonged PR interval in non-LBBB

↓—lower; ↑—higher; CMR—cardiac magnetic resonance imaging; HF—heart failure; HFimpEF—heart failure with improved ejection fraction; LVEDD—left ventricular end-diastolic diameter; AF—atrial fibrillation; AFl—atrial flutter; CRT—cardiac resynchronization therapy; HF—heart failure; LBBB—left bundle branch block; LGE—late gadolinium enhancement; LV GLS—left ventricular global longitudinal strain; TIC—tachycardia-induced cardiomyopathy.

## Data Availability

Not applicable.

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
