# Peer review of "Heart Failure with Improved Ejection Fraction: Insight into the Variable Nature of Left Ventricular Systolic Function"

_ijerph, 2022, doi:10.3390/ijerph192114400_

Round 1

Reviewer 1 Report

line 163, remove the word "its"

line 373 is confusing, ? note versus diuretic should be increased?

line 410 signs rather than sings

well written paper

Author Response

We would like to express our gratitude for the review of the article and the constructive remarks regarding its contents. We trust that, thanks to all the constructive Reviewer’s suggestions, the manuscript has been substantially improved.

  1. line 163, remove the word "its"

Thank you for this remark – we have corrected the manuscript.

  1. line 373 is confusing, ? note versus diuretic should be increased?

We have restructured this sentence so that it is plain to the reader.

  1. line 410 signs rather than sings

We apologize for this shortcoming. We have corrected this expression.

Reviewer 2 Report

This is a well written manuscript dealing with very interesting topic. However, the manuscript is lengthy and difficult to follow due to too many details. I suggest that the overall length of the manuscript should be significantly reduced. Specifically, section 6 functional mitral valve insufficiency  and 6.1 HFimpEF in the context of specific aetiologies of heart failure should be completely omitted, whereas sections 3. General therapy of heart failure irrespective of aetiology and 5. Cardiac implantable electronic devices and HFimpEF should be reduced by at least 40%. I would suggest that the recent findings of Deliver study should be thoroughly discussed with focus on significant number of patients with HFimpEF that were included in the study

Author Response

We would like to thank the Reviewer for all the constructive remarks that helped to improve our manuscript. We have addressed all the issues point by point and provided modifications marked in red.

This is a well written manuscript dealing with very interesting topic. However, the manuscript is lengthy and difficult to follow due to too many details. I suggest that the overall length of the manuscript should be significantly reduced. Specifically, section 6 functional mitral valve insufficiency  and 6.1 HFimpEF in the context of specific aetiologies of heart failure should be completely omitted, whereas sections 3. General therapy of heart failure irrespective of aetiology and 5. Cardiac implantable electronic devices and HFimpEF should be reduced by at least 40%.

We thank the Reviewer for this suggestion. We have considerably shortened the manuscript in the corresponding chapters, i.a. we have deleted Table 2.

 I would suggest that the recent findings of Deliver study should be thoroughly discussed with focus on significant number of patients with HFimpEF that were included in the study

We have added the citation of Deliver study and have provided overview of its results in the context of heart failure with improved ejection fraction. Deliver study was the first randomized controlled trial to include patients with HFimpEF. The results showed that HFimpEF had even greater improvement in terms of primary endpoint than patients without former LVEF <40%.

We have also provided reference to REVIVED-BCIS2 trial in the context of percutaneous revascularization in patients ischaemic aetiology of heart failure.

Reviewer 3 Report

In this manuscript, the authors introduced the definition, epidemiology, predictors, clinical significance of a new sub-group of heart failure with improved ejection fraction (HFiEF). In addition, the authors also discussed the possible principles of therapy for HFiEF. Overall, this is an interesting and comprehensive study, the context was well organized while the abstract may need improvement to help readers to understand.

Specific questions:

1.How about the incidence of HFiEF among all population, all HF patients and CVD high risk population?

2. Is there any genetic association or risk for HEiEF? Given that many genetic associated diseases have links with HEiEF in Table.3.

3. Is there any research model (i.g mice or rat) has been used in HFiEF study? 

Author Response

We would like to express our gratitude for the review of the article and the constructive remarks regarding its contents.

In this manuscript, the authors introduced the definition, epidemiology, predictors, clinical significance of a new sub-group of heart failure with improved ejection fraction (HFiEF). In addition, the authors also discussed the possible principles of therapy for HFiEF. Overall, this is an interesting and comprehensive study, the context was well organized while the abstract may need improvement to help readers to understand.

We have improved the abstract so as to make it more comprehensive to the reader.

Specific questions:

1.How about the incidence of HFiEF among all population, all HF patients and CVD high risk population?

The precise incidence of HFiEF is difficult to assess due to it’s heterogeneous definition and varying aetiologies. Among all HF patients the incidence varies from 14% to 33% (DEVGUN). In population with coronary artery disease (CAD) and HFrEF around 36% of patients fell into JACC’s HFimpEF definition [WILCOX, HAOZHANG HUANG] and they were characterized by lower rate of long-term all-cause mortality. Predictors of HFimpEF in CAD population were related with less advanced progression of disease (e.g. lack of PCI, lack of MI, lower left ventricular end diastolic diameter).

We have referred to the incidence of HFimpEF in the section 2.2:

‘The rate of HFimpEF among the population of patients with HF depends on the applied definition, the proportion of patients with different aetiolo-gies and reversible causes of HF and the intensity and appropriateness of HF therapy. In the Val-HeFT trial, criteria of improvement were met in 9.1% of 3519 patients with baseline LVEF <35% [11]. The longitudinal analysis of trajectory of LVEF variations by Savarese et al. based on Swe-dish Heart Failure Registry showed that 26% with baseline HFrEF and 25% with initial HFmrEF improved to better systolic function subtype of HF [13]. In a recent study by Su and coworkers, HFimpEF defined as in-crease of LVEF by  ≥10% to >40% occurred in 18% of patients [23]. In a recent research by Li et al, HFimpEF defined as an absolute increase of LVEF by 10% (regardless of baseline value) was found in 41.2% of cases [24]. In a recent meta-analysis by He et al per-formed on 9491 from 9 studies, HFimpEF was present in 22.6% of patients [25]. All in all, the prevalence of HFimpEF varies from roughly 10 to 40% [9].’

  1. Is there any genetic association or risk for HEiEF? Given that many genetic associated diseases have links with HEiEF in Table.3.

2According to the JACC Scientific Expert Panel about HFrecEF [WILCOX] a thorough 3-generation family history is always recommended among patients with nonischaemic DCM. So far, confirmed genetic correlations are as follows:

- presence of truncating variants of the titin gene (TTNtv +) are compatible with recovery after introduction of guideline directed medical therapy (DGMT). Moreover, 15% of women with peripartum cardiomyopathy carried truncations in TTN [Ware JS, Seidman JG, Arany Z. Shared genetic predisposition in peripartum and dilated cardiomyopathies. N Engl J Med 2016;374:2601–2.]

- presence of pathogenic mutations in Lamin A/C (LMNA), Desmoplakin (DSP, one of desmosomal gene), SCN5A, and Filamin C (FLNC) confer high risk for sudden cardiac death despite HFrecEF status. However, all-cause mortality was no different between variant carriers and noncarriers [Gigli M, Merlo M, Graw SL, et al. Genetic risk of arrhythmic phenotypes in patients with dilated cardiomyopathy. J Am Coll Cardiol 2019;74: 1480–90.]

  1. Is there any research model (i.g mice or rat) has been used in HFiEF study? 

Recovery of LVEF is linked with the concept of reverse LV remodeling. Cardiac remodeling involves the coordinated regulation of multiple molecular and cellular changes. Biological and theoretical basis for reverse LV remodeling were vastly explored. Weinheimer et al. [Circ Heart Fail 2018;11:e004351.] developed mouse model of HF and provided seminal insights into mechanisms of reverse LV remodeling.

Round 2

Reviewer 2 Report

No further comments